# PLUTO: Pathology-Universal Transformer

Dinkar Juyal [* 1]  Harshith Padigela [* 1]  Chintan Shah [* 1]  Daniel Shenker [1]  Natalia Harguindeguy [1 2]  Yi Liu [1]
Blake Martin [1]  Yibo Zhang [1 2]  Michael Nercessian [1 2]  Miles Markey [1]  Isaac Finberg [1]  Kelsey Luu [1]
Daniel Borders [1]  Syed Ashar Javed [1]  Emma L Krause [1]  Raymond Biju [1]  Aashish Sood [1]  Allen Ma [1 2]
Jackson Nyman [1]  John Shamshoian [1]  Guillaume Chhor [1]  Darpan Sanghavi [1]  Marc Thibault [1]  Limin Yu [1]
Fedaa Najdawi [1]  Jennifer A. Hipp [1]  Darren Fahy [1]  Benjamin Glass [1]  Eric Walk [1]  John Abel [1]
Harsha Vardhan Pokkalla [1 2]  Andrew H. Beck [1]  Sean Grullon [1]

## Abstract

Pathology images provide a unique challenge for computer-vision-based analysis: a single pathology Whole Slide Image (WSI) is gigapixel-sized and often contains hundreds of thousands to millions of objects of interest across multiple resolutions. In this work, we propose PathoLogy Universal TransfOrmer (PLUTO): a light-weight pathology FM that is pre-trained on a diverse dataset of 195 million image tiles collected from multiple sites. We design task-specific adaptation heads that utilize PLUTO's output embeddings for tasks that span pathology scales ranging from subcellular to slide-scale, including instance segmentation, tile classification, and slide-level prediction. We find that PLUTO matches or outperforms existing task-specific baselines and pathology-specific foundation models, some of which use orders-of-magnitude larger datasets and model sizes when compared to PLUTO. Our findings present a path towards a universal embedding to power pathology image analysis, and motivate further exploration around pathology foundation models in terms of data diversity, architectural improvements, sample efficiency, and practical deployability in real-world applications.

## 1. Introduction

### 1.1. Foundation Models in Pathology

Pathology as a medical discipline is instrumental in providing diagnostic and prognostic information to clinicians and patients. In a pathology workflow, surgical tissue specimens are collected, stained, and fixed for microscopy. Microscopic analysis of the tissue is used to establish a diagnosis, estimate disease severity, and identify relevant clinical features for treatment (Walk, 2009; Madabhushi & Lee, 2016; Ehteshami Bejnordi et al., 2017). Each image (WSI or slide) contains up to millions of cells and can be gigapixels in scale, making an exhaustive quantitative manual analysis of WSIs nearly impossible. In addition, information for making pathologic decisions or classifications may exist at multiple scales, from several $\mu$m to several cm, complicating analysis.

Foundation Models (FMs) are promising for pathology as they can take advantage of large amounts of unlabeled data to build rich representations which can be easily adapted for downstream tasks in a data-efficient manner. The diversity of pre-training data results in these models generating robust representations, enabling them to generalize better than individual task-specific models trained on smaller datasets.

Additionally, sharing a backbone across different tasks could also reduce the development and maintenance overhead associated with bespoke task-specific models. Given this prospect, the computational pathology community has made rapid progress in applying self-supervised techniques that have shown promise on natural images such as DINO (Caron et al., 2021), iBOT (Zhou et al., 2021), and DINOv2 (Oquab et al., 2023) to pathology. Most of these efforts have relied on pre-training with a large amount of proprietary data and scaling up the number of backbone parameters used in order to demonstrate high performance on various downstream tasks including tissue classification, disease subtyping classification, and cancer histology segmentation (Kang et al., 2023), (Filiot et al., 2023), (Vorontsov et al., 2023), (Dippel et al., 2024), (Chen et al., 2024).

### 1.2. Our Approach

We designed and built the *PathoLogy-Universal TransfOrmer*, or PLUTO, a state-of-the-art pathology foundation model that, inspired by the dwarf planet, is based on a novel

---

[*]Equal contribution  [1]PathAI, Boston, Massachusetts, USA
[2]Employee of PathAI at the time of study. Correspondence to:
Sean Grullon <sean.grullon@pathai.com>.

*Accepted at the 1st Machine Learning for Life and Material Sciences Workshop at ICML 2024.* Copyright 2024 by the author(s).

light-weight ViT backbone that is pre-trained on a diverse dataset from multiple sites and extracts meaningful representations across the levels of the WSI pyramid outlined in figure 1. The key features of PLUTO are outlined below:

1. **Pre-training Dataset** We compiled a large dataset across a diverse spectrum of histology stains, scanners, and biological objects across resolution scales that include 200+ biologically-meaningful objects and region types (which we term *substances*) from more than 50 sources (Section 2.1).

2. **Architecture** We designed the PLUTO backbone to generate informative feature representations at different length scales from a compact ViT backbone. We achieved this by implementing a self-supervised learning scheme that accommodates flexible patch sizes from the FlexiViT scheme (Beyer et al., 2023), extending it to accommodate multiple magnifications during training, and modifying the DINOv2 loss by adding a Masked Autoencoder (MAE) (He et al., 2022) objective and a Fourier-loss-based term to modulate the preservation of low- and high-frequency components (Section 2.2).

3. **Multi-scale Evaluation** We evaluated the quality of the resulting FM by constructing a suite of adaptation heads to perform diverse, challenging tasks across the levels of the WSI pyramid, and evaluated performance across different biologically-relevant benchmarks (Section 3).

4. **Deployability.** Performing a computational pathology task may require embedding tens to hundreds of thousands of WSI tiles to make a single prediction. To enable this, we focused on developing a model that was efficient (Section 3.3).

## 2. Methods

### 2.1. Pre-training Data Characteristics

The dataset used for self-supervised pre-training comprises public and proprietary datasets, totaling 195M image tiles sampled at four resolutions from $158,852$ WSIs derived from over 50 source site, 28 disease areas, 12 scanners and 200 substances. Following findings from DINOv2 (Oquab et al., 2023) highlighting the significant value of incorporating curated data into self-supervised pre-training, this dataset is augmented with an additional set of samples extracted from over four million manual annotations from board-certified pathologists. During pre-training, the labels are discarded, but the inclusion of pathologist-curated regions covering a wide range of biological patterns provides an implicit data diversity in the pre-training process.

This source of biological diversity, combined with the broad range of stains, organs, diseases, and source sites, makes this one of the most diverse large-scale digital pathology datasets to date.

### 2.2. PLUTO Architecture Overview

PLUTO (Figure 1) is designed with specific characteristics in mind to enable its usage on a wide range of use-cases, as described in Section 1.2. To design the PLUTO architecture, we start from DINOv2 (Oquab et al., 2023) which combines DINO (Caron et al., 2021) and iBOT (Zhou et al., 2021) losses (along with KoLeo regularizer) to learn relevant representations at the tile and patch levels respectively. Note, we use **tile** and **image** interchangeably to refer to the image tiles and **patches** to refer to the patch-tokens obtained by dividing the tile into smaller patches for processing in ViTs. The DINOv2 architecture was primarily developed for natural images that are often object-centric. Pathology WSI tiles on the other hand have thousands of objects such as nuclei, cells, and glands with different sizes, observed at different image resolutions. To design an encoder which can capture details of objects at different levels of granularity, we add in a MAE (He et al., 2022) objective with multi-scale masking. The MAE setup tries to reconstruct masked regions of the input image (often a large fraction of the input) from the unmasked regions. We perform masking by varying the patch sizes used for masking while using images across different resolutions of the WSI as shown in Figure 1. In addition to the pixel-level reconstruction loss used in MAE, we add a Fourier reconstruction loss to control the amount of low- and high-frequency information preserved during the pre-training process.

To enable the encoder and decoder to handle varying patch sizes for multi-scale masking, we employ the FlexiViT setup (Beyer et al., 2023). Since patch size controls the granularity of information captured by the encoder, different downstream tasks may need different patch sizes for optimal performance. The FlexiViT setup allows us to adapt the same backbone to different tasks without needing to train a backbone for every patch size. The patch size also determines the effective sequence length used in ViTs and FlexiViT allows us to cater to different compute budgets by selecting the most suitable patch size at inference time.

We observe slightly better performance with the teacher over the student, and thus use the teacher for all downstream tasks. We use ViT-S for the student and teacher encoders, and a shallower model is used for the MAE decoder. For training, we use AdamW with a base learning rate of 0.002 and a learning rate warmup for the first 5 epochs. We use a distributed training setup to scale the training across 64 NVIDIA A40 GPUs.

## A. PLUTO framework for multi-resolution pathology tasks

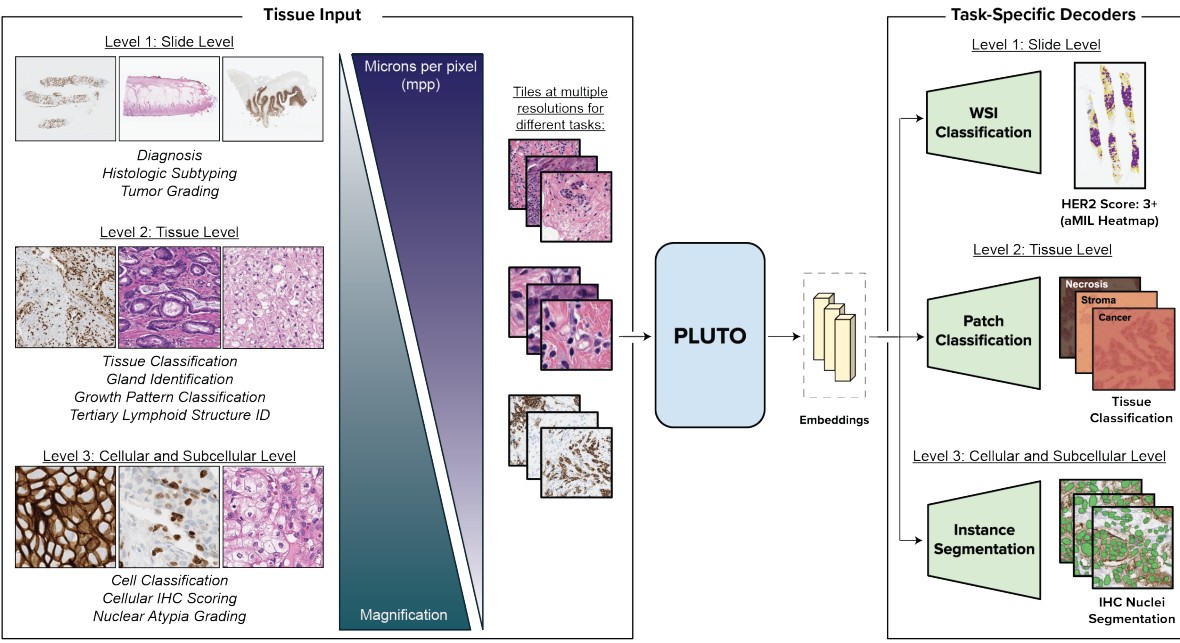

## B. PLUTO training paradigm

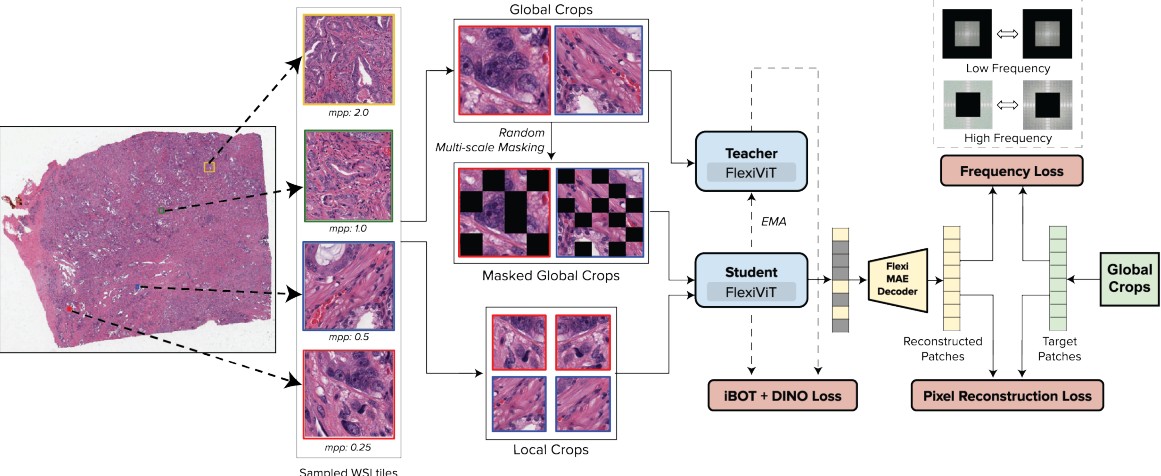

*Figure 1.* Overview of PLUTO. Panel A) outlines the PLUTO multi-resolution adaptation pipeline. Tiles are extracted from WSIs at multiple resolutions and correspond to scales that capture different biological contexts. We organize pathology tasks according to these biological contexts as slide level, tissue level, and cellular & subcellular level tasks, respectively. PLUTO generates embeddings that are task-agnostic and can be used in a variety of downstream tasks, where adaptation to WSI-level prediction, tile classification, and instance segmentation are shown. Panel B) demonstrates the PLUTO architecture in detail. WSI tiles at multiple resolutions are masked with varying patch sizes and passed to the backbone for self-supervised pre-training. The architecture is optimized for flexibility across multiple scales and patch sizes. In addition to DINO and iBOT losses, MAE and Fourier losses are applied across varying mask sizes to control the amount of low- and high-frequency information that is preserved.

*Table 1.* Performance of MIL models with different ViT- and CNN-based featurizers on NSCLC subtyping task. The mean and standard deviation across 1, 000 bootstrapped runs are reported. We note that MIL models that use our frozen PLUTO model as a featurizer tend to outperform models with both frozen and fine-tuned CNN backbones (ShuffleNet) and Imagenet-pre-trained ViT backbones. This is especially evident in OOD performance, highlighting the robustness of PLUTO's embeddings.

| Model | Dataset | Patch Size | Tuning | In-domain F1 | In-domain AUROC | OOD F1 | OOD AUROC |
|---|---|---|---|---|---|---|---|
| PLUTO | NSCLC | 16 | Frozen | **90.2(1.9)** | **94.0(1.6)** | **86.1(2.8)** | **91.2(2.5)** |
| Meta-DINOv2 ViT-S | NSCLC | 14 | Frozen | 88.6(2.0) | 92.0(1.9) | 72.1(4.1) | 81.9(3.8) |
| ShuffleNet | NSCLC | - | Frozen | 83.6(2.4) | 90.1(2.0) | 72.2(4.2) | 83.5(3.5) |
| ShuffleNet | NSCLC | - | Fine-tuned | 88.1(2.2) | 93.9(1.5) | 42.5(8.0) | 90.8(2.1) |

*Table 2.* Summary of PLUTO performance across tile classification and segmentation datasets. We perform the tile classification task on CRC-100K and Camelyon17-WILDS datasets by linear probing the PLUTO embeddings while keeping the backbone frozen. PLUTO, while significantly smaller, achieves strong performance and is competitive with the best performing models that have been reported for these two datasets, highlighting the effectiveness of diverse pre-training data for enhancing robustness. We perform the gland segmentation and nuclei segmentation tasks on the GlaS and PanNuke datasets, respectively, by adapting PLUTO through multiple adaptation strategies while keeping the backbone frozen. PLUTO achieves state-of-the-art performance on gland segmentation, outperforming other fully supervised segmentation frameworks. PLUTO beats fine-tuned backbone baselines of comparable size on the PanNuke dataset, and is competitive with significantly larger fine-tuned backbones that have been reported for PanNuke.

| Model | Adaptation Head | Benchmark Name | Metrics | |
|---|---|---|---|---|
| | | | Acc. | Bal. Acc. |
| **PLUTO** | **Linear Head** | **H&E CRC-100K** | **96.6** | **95.3** |
| ResNet50* | N/A | H&E CRC-100K | 94.7 | N/A |
| | | | Acc. | Bal. Acc. |
| **PLUTO** | **Linear Head** | **Camelyon17-WILDS** | **96.2** | - |
| DenseNet-121* | N/A | Camelyon17-WILDS | 70.3 | - |
| | | | DICE | IoU |
| **PLUTO** | **Mask2Former** | **GlaS** | **91.2** | **84.5** |
| PLUTO | Mask R-CNN | GlaS | 88.0 | 79.6 |
| UNet* | N/A | GlaS | 85.5 | 74.8 |
| | | | bPQ | mPQ |
| **PLUTO** | **HoverNet** | **PanNuke** | **67.1** | **47.7** |
| PLUTO | Mask R-CNN | PanNuke | 58.6 | - |
| ResNet50 + Mask R-CNN* (Shui et al., 2023) | N/A | PanNuke | 55.3 | 36.9 |

*Fully Supervised Baseline Model

## 2.3. PLUTO Adaptation

The backbone training process outlined above learns generic, task- features. In order to leverage its general capabilities, we add task-specific heads and *adapt* these heads through supervised fine-tuning, while keeping the backbone fixed, or frozen. This adaptation process is efficient and provides the flexibility to use the same pre-trained backbone for specialized tasks across the biological scales. Although different tasks may require the use of different patch sizes to capture relevant context, the FlexiVit setup allows us to dynamically select the backbone patch size for adaptation.

### 2.3.1. SLIDE-LEVEL TASK ADAPTATION

We adapt PLUTO to Level 1 slide-level tasks by performing weak supervision on slide-level labels. In particular, MIL (Ilse et al., 2018) is a weakly supervised learning technique where sets of instances are grouped into a "bag" and used to learn bag-level labels. These MIL models consist of three parts: (1) a featurizer which generates representations of each image tile in a bag, (2) an aggregation module which combines tile representations using a permutation-invariant function (typically attention) to generate a bag-level representation, and (3) a classifier which outputs a bag-level prediction. We adapt our FM backbones by using the pre-trained backbones directly as featurizers. We use the AdditiveMIL classifier (Javed et al., 2022), which enables interpretable model predictions and class-wise heatmaps.

### 2.3.2. TISSUE-LEVEL, CELLULAR- AND SUBCELLULAR-LEVEL TASK ADAPTATION

We adapt PLUTO to tissue-level and cellular/subcellular-level biological scales through fine-tuning either a tile classification or an instance segmentation adaptation head. These two adaptation strategies are informed by the availability of labeled data: tile-level classification only requires labels at the image tile level, whereas instance segmentation requires pixel-level annotations.

We explore and benchmark a range of adaptation head architectures for tile classification, ranging from single linear layers to multilayer perceptrons (MLPs) with different pooling strategies.

We adapt the SSL-pre-trained ViT backbone to instance segmentation tasks via two distinct frameworks: Mask R-CNN (He et al., 2017) and Mask2Former (Cheng et al., 2021). To the best of our knowledge, this is the first work comparing Mask2Former to ViT + Mask R-CNN approaches on histopathology tasks. We also experimented with combining the ViT with a ViT-Adapter (Chen et al., 2022), which has been shown to improve segmentation performance. The output feature maps of the adapter, corresponding to different spatial resolutions of the input image, are used as the input

to Mask R-CNN and Mask2Former.

# 3. Results

## 3.1. Slide-level prediction

We consider the prediction of the cancer subtypes Adenocarcinoma and Squamous cell carcinoma in non-small cell lung carcinoma (NSCLC) H&E-stained WSIs, a popular benchmark for slide-level evaluation. For NSCLC subtyping, we use slides from the publicly available TCGA Adenocarcinoma (LUAD) and Squamous Cell Carcinoma (LUSC) groups. We use 500 slides for model development and 247 (128 LUAD / 119 LUSC) slides for test set evaluation. We evaluate out-of-distribution (OOD) performance using a proprietary dataset of 205 WSIs (162 Adenocarcinoma WSIs, 45 Squamous Cell Carcinoma WSIs) collected from a different source site with varying image acquisition and processing steps, resulting in visual differences from the TCGA WSIs. Results are shown in Table 1, where PLUTO improves upon other models on ID and OOD sets.

## 3.2. Tile Classification and Instance Segmentation

We used two publicly available datasets: CRC-100K (Kather et al., 2018) and Camelyon17-WILDS (Bandi et al., 2019; Koh et al., 2021). The CRC-100K dataset consists of $107,180$ images ($224 \times 224$ at $0.5$ mpp) of human colorectal cancer (CRC) and normal tissue extracted from 136 H&E histopathology WSIs classified into one of nine tissue classes. The Camelyon17-WILDS dataset contains $455,954$ images ($96 \times 96$ pixels at 1 mpp, downsampled from $0.25$ mpp slides) from 50 WSIs of breast cancer metastases in lymph node sections from five different hospitals. The task is a binary classification of whether the central $32 \times 32$ region contains tumor tissue. For instance segmentation, we use 2 public datasets. First is PanNuke (Gamper et al., 2019) that 481 visual fields across 19 different tissue types from WSIs from TCGA and a local hospital, with a total of $189,744$ exhaustive nuclei labels categorized into five classes. Second dataset is GlaS (Sirinukunwattana et al., 2017) which consists of 85 gland morphology images for training and 80 images for testing. Results in table 2 demonstrate that PLUTO matches or outperforms task-specific baselines (Nguyen et al., 2024) for these datasets.

## 3.3. Deployability

To illustrate the efficiency of PLUTO, we compare the throughput efficiency of various ViT backbones (ViT-S, ViTB, ViT-L, ViT-H) for two common pathology tasks: tile classification and slide-level prediction. We note that we have not applied any inference-specific optimizations in this setup. We use the same data-loading pipeline and hardware (A40 GPU) for all the backbones. As seen in Figure 2, for

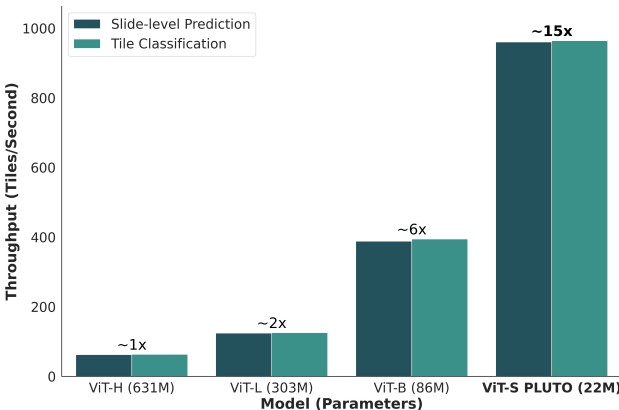

*Figure 2.* Throughput (tiles/sec) of models for tile-level and slide-level classification tasks with various backbones using patch size 16 with a tile size 224 × 224. We use linear probes and AdditiveMIL classifiers as adaptation heads respectively for the tile and slide-level classification tasks. Notable pathology FMs use ViT-H (Vorontsov et al., 2023), ViT-L (Chen et al., 2024) (Dippel et al., 2024) and ViT-B (Filiot et al., 2023).

both the tasks, ViT-S is around $2.5\times$ faster than ViT-B, $7.5\times$ faster than ViT-L, and $15\times$ faster than ViT-H.

## 4. Conclusion

We present in this paper PLUTO: a competitive state-of-the-art pathology Foundation Model based on a light-weight ViT. PLUTO is designed to take advantage of the multi-scale nature of WSIs and provide informative representations across biological scales. We have quantified the performance of PLUTO on a variety of adaptation tasks across biological scales. Our work also demonstrates the importance of incorporation of biological priors in the construction of pre-training datasets and the design of the model architecture for large-scale self-supervised models. We hope that our efforts with PLUTO further motivate building high-performing, deployable FMs; drive FM adoption in pathology; and serve real-world translational research and clinical applications.

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
