# Supplementary Section for PLUTO: Pathology-Universal Transformer

## 1. Pyramid Structure of Whole Slide Images

WSIs are digitized and stored in a multi-scale pyramidal structure, where the base of the pyramid is the highest-resolution image data as captured by the slide scanner. The resulting scan of a WSI can reach $200,000 \times 200,000$ pixels at a full resolution of $0.25$ microns ($\mu$m) per pixel (mpp) (Sellaro et al., 2013); however, different "levels" of the pyramid may be accessed for different purposes.

Biological entities observed on WSIs vary dramatically in scale, and therefore pathologists will commonly move between magnifications to assess different aspects of a tissue sample on a pathology slide (Molin et al., 2016). At low magnification, pathologists may scan across slides to identify regions of interest in the tissue, with characteristic lengths of approximately 1 mm–1 cm. At middle magnification (such as $5$–$10\times$) pathologists commonly view structures at length scales of $200$ $\mu$m–1 mm. At this scale, pathologists distinguish between tissue types, glands, tumor growth patterns, histologic subtypes of diseases, or other multicellular entities in the image. At high magnification ($20$–$40\times$) it is possible to resolve entities $1$ $\mu$m–$50$ $\mu$m in length, such as individual cell identities, subcellular structural morphology used in determining malignancy, or localization of immunohistochemical (IHC) staining (Magaki et al., 2019).

The hierarchical nature of biological entities necessitates considering the multiple scales at which information must be extracted and used by ML algorithms. For example, passing a $224 \times 224$ image tile at $0.25$ mpp through an encoder developed for encoding at 1 mpp may completely miss relevant nuclear pleomorphism, whereas passing a $224 \times 224$ tile at 1 mpp through an encoder developed for encoding at $0.25$ mpp may be unable to adequately distinguish between acinar and lepidic growth patterns. For clarity, we organize pathology tasks according to such biological scales as follows:

- **Level 1: Slide Level** This scale includes tasks that label the entire slide such as predicting driver gene mutations in cancer, histologic subtyping, or tumor grading. However, it is uncommon that slide-level assessments are made at slide-level magnification. Typically, assessments made at this scale are aggregated across evaluation of higher-magnification tiles.

- **Level 2: Tissue Level** This is the scale at which it is possible to identify and characterize tissue regions (e.g. cancer regions and necrotic regions) and many-cellular objects such as glands.

- **Level 3: Cellular and Subcellular Level** This is typically the maximal resolution of a WSI, where cellular and subcellular morphology is evident.

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

*Figure 1.* Dataset characterization for the pre-training dataset. The distribution of the dataset by organ, disease, stain, scanner, and objective magnification is shown, as well as the distribution of cell point and tissue region annotations which augment the pre-training dataset (NOS: Not Otherwise Specified). Aggregate data characteristics are summarized above these distributions which also indicate the number of biologically-meaningful objects and region types, which we term *substances* (e.g. lymphocyte, blood vessel, Gleason pattern 3 prostate cancer, tumor bed). The large number of source sites (50+) guarantees large diversity during PLUTO self-supervised pre-training.