# OpenReview forum: "PLUTO: Pathology-Universal Transformer"
_ICML.cc/2024/Workshop/ML4LMS — ML4LMS Poster_

### Official Review · Reviewer_mNBx · 2024-06-11
**Good paper with clear contribution and novelty**

**Rating:** 8
**Confidence:** 4

**Review:**

The authors provide a well-presented review of relevant work on foundation/ self-supervised models in imaging, particularly challenges in pathology. References to existing FM in pathology would be useful.

Easy-to-read structure with clear contribution across multiple impact areas

The authors do a good job highlighting the limitations of DINO2 in capturing representations of pathological images and the steps to address those limitations.

The adaptation of the task-specific heads using supervised fine-tuning with a FlexiVit setup is well presented and explains some of the efficiency gains.

Given the diversity of histopathological conditions evaluation on OOD dataset is crucial to assess generalizibility of the model. It would be helpful if authors whawould hint a bit more about the expected visual differences in this distribution to understand the limitations of this evaluation.

---

### Official Review · Reviewer_VQcy · 2024-06-12
**Review of PLUTO: Pathology-Universal Transformer**

**Rating:** 9
**Confidence:** 5

**Review:**

This paper proposes a new pathology foundation model (FM) with task-specific adaptation heads, applicable at the slide-level to the subcellular level. The authors aim to create an embedding that can be used at multiple scales for multiple different tasks on pathology images, including segmentation, classification and prediction.

To develop good FMs, a large and diverse dataset is crucial. Here, the authors have used a very diverse dataset of 195 million tiles from 158,852 whole slide images, from 12 different scanners at four resolutions and over 28 disease areas. The authors chose to include high quality pathologist-curated data in the training set, as opposed to using this data as the testing set, with the aim of optimising the use of this data for broader applications. The authors claim to have a diverse spectrum of histology stains in their dataset, but no further details are provided on what stains are used.

The design choices are well motivated, such as using a MAE because DINOv2 alone is designed for object-centric natural images, not heterogenous pathology images. A lightweight Vision Transformer is also an appropriate option, since there is likely enough data to feed this model which, through its self-attention mechanism, considers the context of the surrounding tissue which is valuable for clinical inference on pathology. The choice of position embedding in the ViT architecture has not been detailed but would be informative for model understanding and transparency. The use of a Fourier-based reconstruction loss term seems like a reasonable design choice here, and an ablation study on the presence of this term would be interesting (but not necessary for the scope of this paper). For slide-level inference the AdditiveMIL classifier is used. Linear layers and MLPs with various pooling strategies are explored as the adaptation heads, and the choice is left up to the user, depending on the task.

The FM is adapted and tested on a slide-level cancer subtype prediction task on TCGA NSCLC data; tile tumour classification on Camelyon-17-WILDS  and CRC-100K; and instance segmentation of nuclei on PanNuke and glands on GlaS. These tasks are not considered challenging within the field, yet demonstrate the use of the proposed FM for tasks at different scales. The GlaS segmentation example in particular demonstrates the application of the FM with limited training data for the adaptation heads, and the classification results from using only a linear layer as the adaptation head demonstrate the quality of the feature embeddings.

It would be interesting, and important for understanding the generalisability of this work, to see how the FM applies to more complex tasks, potentially also with limited data for training the adaptation heads. The performance of the FM is compared to that of fully supervised baseline models or ImageNet pre-trained models, but not to other pathology FMs, of which many are publicly available. I do however acknowledge that the number of results that can be shown is somewhat limited by the length of the paper in this workshop.
Section 3.2 is not written consistently and should be made clearer e.g. “First is PanNuke (Gamper et al., 2019) that 481 visual fields across 19 different tissue type … ”. It’s not explicitly stated what the segmentation task is here, but from the context the reader may assume it to be segmenting nuclei and glands for the respective datasets, and the details are later provided in the caption of Table 2.

Notably, the authors have access to a huge amount of compute, allowing them to train such a model using 64 A40 GPUs. The computing resources available to others wanting to use this FM has been considered in the choice of model architecture. FlexiViT can dynamically select the most suitable patch size in Vision Transformer (what the VIT breaks down the image tile further into for calculation of self-attention) at inference time, making it flexible to computing capacity and image size.

FMs are generally developed with direct translatability in mind, to be used by fellow researchers and/or commercial clients. This work is no different and could be a useful resource to researchers exploring a wide range of pathology inference tasks at various scales.